# Factors Associated with Deterioration of Primary Angle Closure after Lens Extraction

**DOI:** 10.3390/jcm11092557

**Published:** 2022-05-02

**Authors:** Min Kyung Song, Joong Won Shin, Kyung Rim Sung

**Affiliations:** 1Department of Ophthalmology, Ilsan Paik Hospital, College of Medicine, Inje University, Goyang 10380, Korea; maban39@daum.net; 2Department of Ophthalmology, Asan Medical Center, College of Medicine, University of Ulsan, Seoul 05505, Korea; sideral@hanmail.net

**Keywords:** lens extraction, primary angle closure glaucoma, progression, risk factor

## Abstract

The purpose of the study was to explore factors associated with glaucomatous deterioration in eyes with primary angle closure (PAC) after lens extraction, including PAC suspect (PACS), PAC, and PAC glaucoma (PACG). We retrospectively analyzed data of 77 eyes with PACS, PAC, and PACG that underwent lens extraction with more than 2 years postoperative follow-up. Postoperative glaucoma progression was analyzed by either structural (optic disc/retinal nerve fiber layer (RNFL) photographs or optical coherent tomography (OCT)) or functional (visual field (VF)) criterion. Cox proportional hazard analysis (hazard ratio (HR)) was used to determine risk factors for progression using uni-and multivariate analysis. The analysis was conducted in groups with or without glaucomatous optic neuropathy (PACS/PAC vs. PACG). Forty-one eyes with PACS/PAC and 36 eyes with PACG were included. The mean postoperative follow-up period was 3.5 ± 1.4 years. Intraocular pressure (IOP) was reduced postoperatively from 23.1 ± 14.4 to 13.4 ± 2.1 mmHg. In the PACS/PAC group, seven eyes (17.0%) showed structural progression, but none showed progression in VF. Preoperative RNFL thickness was the only risk factor for structural progression (HR = 0.928, *p* = 0.002) in the PACS/PAC group. In the PACG group, 24 eyes (66.7%) showed structural progression and 12 eyes (33.3%) showed VF progression. Thinner preoperative RNFL thickness (HR = 0.964, *p* = 0.043) and high postoperative IOP fluctuation (HR = 1.296, *p* = 0.011) were significantly associated with VF progression; none of the factors were associated with structural progression. Angle closure eyes with thinner baseline RNFL thickness and higher postoperative IOP fluctuation may require careful follow-up for glaucoma progression after lens extraction.

## 1. Introduction

Primary angle closure glaucoma (PACG) has a generally worse prognosis than primary open angle glaucoma [1,2]. PACG is a type of glaucoma associated with iridotrabecular contact (ITC) on gonioscopic examination, which leads to raised intraocular pressure (IOP) and results in glaucomatous optic neuropathy. Primary angle closure suspect (PACS) and primary angle closure (PAC) are a spectrum of primary angle closure diseases (PACD) with narrow or occludable angle causing a predisposition to PACG [3]. Treatments for PACD include surgical interventions such as iridoplasty, iridotomy, and lens extraction that open the angle to resolve ITC. Among these treatments, lens extraction has proven that it can relieve pupillary block, deepen the anterior chamber, and resolve some of the adhesion of the anterior chamber angle; consequently, it is a more effective treatment for PACD. Significant IOP reduction was also achieved after lens extraction [4,5,6,7,8,9,10]. However, despite IOP control, the surgical results are unsatisfactory in some patients with PACG who required subsequent filtration surgery or more antiglaucoma drugs [9,11,12]. In our previous study, despite well controlled IOP, a substantial proportion of patients with PACG showed structural and visual field progression. Eyes with PACS and PAC also showed structural progression. Additionally, there was a significant difference in rates of progression in PACG compared with the PACS and PAC groups [13]. Hence, we aimed to explore the specific factors associated with glaucomatous deterioration in eyes with a spectrum of PACD after lens extraction.

## 2. Materials and Methods

We consecutively enrolled patients with PACS, PAC, and PACG who had been diagnosed at the glaucoma clinic of the Asan Medical Center and underwent lens extraction by phacoemulsification with intraocular lens implantation by a single surgeon (KRS) from March 2008 to April 2017. All subjects underwent a complete ophthalmic examination, including best-corrected visual acuity (BCVA), slit-lamp examination, Goldmann applanation tonometry, gonioscopy, fundoscopy, stereoscopic optic disc photography, retinal nerve fiber layer (RNFL) photography, axial length (AL) measurement (IOLMaster; Carl Zeiss Meditec, Dublin, CA, USA), visual field (VF) test (Humphrey field analyzer (HFA), Swedish Interactive Threshold Algorithm (SITA) 24-2, Carl Zeiss Meditec, Dublin, CA, USA), optic nerve head imaging with spectral-domain optical coherence tomography (SD-OCT, Cirrus HD-OCT, Carl Zeiss Meditec), and anterior segment OCT (AS OCT, Visante OCT, Carl Zeiss Meditec). The central corneal thickness (CCT) was measured using AS OCT. The study was approved by the Institutional Review Board of the Asan Medical Center, and the tenets of the Declaration of Helsinki were followed.

Patients were grouped based on the ISGEO (International Society of Geographical and Epidemiological Ophthalmology) criteria [3]. Eyes with PACS were defined in the static gonioscopic examination as having appositional contact >270° between the peripheral iris and the posterior trabecular meshwork. The PAC group included eyes with occludable angles and had features indicating trabecular obstruction by the peripheral iris. Such features included elevated IOP, iris whorling (distortion of the radially orientated iris fibers), “glaucomflecken” lens opacities, excessive pigment deposition on the trabecular surface, or presence of peripheral anterior synechiae (PAS) but without the development of a glaucomatous optic disc or any VF change. PAC eyes showing glaucomatous optic disc changes (neuroretinal rim thinning, disc excavation, and/or optic disc hemorrhage attributable to glaucoma) or a glaucomatous VF change were considered to have PACG. Indentation gonioscopy was performed in all eyes to determine if the anterior chamber angle closure was due to apposition or PAS. Patients who underwent laser peripheral iridotomy (LPI) before or during follow-up were included in the study. We excluded patients with any history or current use of topical or systemic medication that could affect the angle or pupillary reflex, and those with a history of previous intraocular surgery. In addition, patients who underwent lens extraction combined with glaucoma filtering or drainage device implantation were excluded. Eyes diagnosed with secondary angle closure, such as neovascular or uveitic glaucoma, were also excluded. To be included in this analysis, all patients needed to have received follow-up examinations for ≥2 years after lens extraction. If both eyes were eligible, one eye was selected at random. All patients underwent follow-up examinations at 6- to 12-month intervals using stereoscopic optic disc/RNFL photography, VF testing, and SD-OCT scanning.

Lens extraction was performed at the surgeon’s discretion on a patient who agreed to undergo the surgical procedure. Lens extraction was determined based on at least one of the following criteria: (1) shallowing of the anterior chamber, (2) progressive cataract or reduction of visual acuity, (3) inadequate IOP control, and (4) patients’ willingness to undergo the surgery. All surgical procedures were performed uneventfully by a single surgeon (KRS).

An independent examiner performed AS OCT of all eyes in constant dim light (0.5 cd/m^2^) with the patient in a sitting position. A cross-sectional horizontal scan (3- and 9-o’clock meridians; nasal–temporal angles at 0° to 180°) was obtained for each subject. Images with good central fixation, high resolution of the scleral spur, and no motion artifact were selected for analysis. Anterior segment parameters were measured by a masked technician who was blinded to all other test results and to the clinical information of the participants. Various angle parameters, including angle opening distance at 500 μm from scleral spur (AOD500), anterior chamber depth (ACD), and lens vault (LV) were provided by the manufacturer of AS OCT. The AOD500 was defined as the linear distance between the point of the inner corneoscleral wall (500 μm anterior to the scleral spur) and the iris. The average of AOD500 at 180° and 0° was used. ACD was defined as the distance from the corneal endothelium to the anterior surface of the lens. LV was defined as the perpendicular distance between the anterior pole of the crystalline lens and the horizontal line joining the two scleral spurs [14].

The progression of the optic disc and RNFL defects were determined by evaluating the entire series of stereoscopic optic disc/RNFL photographs. Two glaucoma experts (KRS and MKS) independently assessed all photographs to estimate glaucoma progression. In each patient, the most recent photograph was compared with the baseline photograph. The two graders were unaware of the progression assessments made by the other, and each grader viewed all photographs of each eye before making an assessment. Both graders were asked to determine the glaucomatous optic disc or RNFL progression as indicated by an increasing pallor in the optic disc, increase in the extent of neuroretinal rim thinning, enhancement of disc excavation, and/or any widening, deepening, or appearance of new RNFL defects. The graders classified each glaucomatous eye as either stable or progressing. If the two experts had conflicting results, a third examiner (JWS) made the final decision.

SD-OCT images (Cirrus HD-OCT; Carl Zeiss Meditec) were measured at the same visit, as well as the clinical examination and VF test. A detailed description of the principles of SD-OCT has been published previously [15]. Cirrus OCT Guided Progression Algorithm (GPA) software was used to analyze the data. If either optic disc/RNFL or OCT progression was confirmed, it was considered as structural progression.

VF tests were performed using HFA (SITA 24-2; Carl Zeiss Meditec). Only reliable VF test results (false-positive errors < 15%, false-negative errors < 15%, and fixation loss < 20%) were included in the analysis. Glaucomatous VF defect was determined by at least two reliable VF examinations and defined based on the following: those with a cluster of three points with probabilities < 5% on the pattern deviation map in at least one hemifield, including one point with a probability < 1%; glaucoma hemifield test result outside normal limits; or a pattern standard deviation outside 95% of the normal limits [16]. For confirmation, the VF test was repeated within 2 weeks from the baseline measurement. For inclusion in the study cohort, ≥5 reliable VF tests after lens extraction were required. VF progression was determined using both event-based and trend-based approaches. For the event-based analysis, a commercial software program (HFA GPA; Carl Zeiss Meditec) was used. For the GPA, VF progression was defined as a significant deterioration from the baseline pattern deviation at ≥3 of the same test points that were evaluated on three consecutive examinations [17]. For the trend-based analysis, a linear regression analysis using VF mean deviations (MDs) was employed. We defined VF progression as a significantly negative slope (*p* < 0.05). Participants were diagnosed with VF progression when an eye demonstrated progression according to either of these two methods.

Data analysis was conducted using the Statistical Package for Social Sciences version 22.0 (SPSS, IBM Corp., Armonk, NY, USA). Continuous variables were expressed as means ± standard deviations (SDs) after confirming the normality of the data distribution. Postoperative IOP fluctuation was defined as the difference between the peak and trough value of postoperative IOP between each visit. Baseline ocular parameters and clinical data were compared among the two groups with and without glaucomatous optic neuropathy: PACS/PAC and PACG. Comparisons were performed using an unpaired *t*-test, Mann–Whitney test, and chi-square test as appropriate. Hazard ratios (HR) for the association between clinical factors and glaucoma progression (structural and VF) in each group were determined using Cox proportional hazard models. Univariate Cox proportional hazards analyses were performed separately for all putative variables. Factors with a *p* of less than 0.1 in the univariate model were included as independent variables in the multivariate model. A multivariate model using a backward elimination approach, with a likelihood ratio, was created. A *p* value of less than 0.05 was considered significant.

## 3. Results

Among 211 PACS/PAC/PACG patients who underwent lens extraction and were assessed for eligibility, 77 patients (36.4%) fulfilled the criteria of having more than five reliable VFs and at least 2 years of follow-up after lens extraction. Among the subjects, 17 eyes were PACS, 24 were PAC, and 36 eyes were PACG. The demographic and ocular characteristics of all subjects are presented in Table 1. The average age of the total subjects was 66.0 ± 7.6 years, and all subjects were Korean. There were no significant differences between the PACS/PAC and PACG groups in terms of age, gender, AL, CCT, and AS OCT parameters. Baseline IOP was also similar between the two groups. As expected, the PACG group showed lower RNFL thickness (*p* < 0.001), worse VF MD (*p* < 0.001) and visual field index (VFI) (*p* < 0.001) at both preoperative and postoperative exams than the PACS/PAC group. The period between baseline examination and lens extraction was not different between the two groups. Postoperative mean IOP was reduced from baseline to 13.4 mmHg in the PACS/PAC and 13.5 mmHg in the PACG group, and there was no difference between the two groups. The number of glaucoma medications used postoperatively was 0.15 and 0.86 in each group, which was significantly different (*p* < 0.001). The prevalence of glaucomatous progression according to two criteria (structure and VF) differed significantly between groups (*p* < 0.001). In the PACS/PAC group, seven eyes (17.0%) progressed by structural assessment, but none of the eyes (0%) showed VF progression. Among 36 eyes with PACG, 24 eyes (66.7%) showed progression based on structural assessment and 12 eyes (33.3%) showed VF progression.

The clinical data comparison between progressed and non-progressed eyes is shown in Table 2 for the PACS/PAC group, and in Table 3 for the PACG group. In the PACS/PAC group, CCT and preoperative RNFL thickness were significantly lower in progressed eyes (*p* = 0.049 and *p* = 0.001, respectively). In the PACG group, the structurally progressed group showed lower CCT than non-progressed eyes (*p* = 0.041). For VF progression, the progressor showed lower CCT (*p* = 0.026), lower preoperative RNFL thickness (*p* = 0.028), lower preoperative VFI (*p* = 0.024), and larger postoperative IOP fluctuation (*p* = 0.013).

Table 4, Table 5 and Table 6 show the results of Cox regression analyses assessing clinical variables associated with progression in each group according to either structural or functional criteria. For structural progression in the PACS/PAC group (Table 4), univariate and multivariate analysis showed that lower preoperative RNFL thickness was a risk factor (HR = 0.928, *p* = 0.002 in multivariate analysis). In PACG for structural progression (Table 5), there was no significant risk factor for progression. In Table 6, multivariate Cox regression analyses showed that lower preoperative RNFL thickness (HR = 0.964, *p* = 0.043) and higher postoperative IOP fluctuation (HR = 1.296, *p* = 0.011) were independently associated with VF progression.

## 4. Discussion

In the present study, preoperative thinner RNFL was associated with structural progression in PACS/PAC. In PACG, thinner preoperative RNFL and postoperative IOP fluctuation were associated with VF progression. Hence, worse baseline glaucoma severity and inadequate IOP control affected long-term prognosis in the PAC spectrum, even after lens extraction.

Cataract, or clear lens extraction has been suggested as a treatment option for PACD, since it may reduce the risk of glaucomatous progression by opening the anterior chamber angle and, thus, controlling the IOP. Several studies have shown that lens extraction provides an opportunity to restore vision and significantly reduce IOP in PACD [6,7,8,9,10,11,18,19,20]. Especially, the EAGLE study emphasized the effectiveness of early lens extraction for the treatment of PACG in a randomized controlled trial [6].

However, lens extraction itself in patients with PACG is not always sufficient to prevent the progression of glaucoma. Previously, in patients with medically controlled chronic angle closure glaucoma (CACG), 11.4% showed glaucomatous progression in their optic disc and 14.3% showed progression in VF after two years of phacoemulsification. Regarding patients with medically uncontrolled CACG, 40.7% showed VF progression after 2 years of follow-up [21,22]. A recently published five-year follow-up study for these groups reported that 43.3% of CACG showed optic disc progression, and 18.8% of patients showed VF progression [9]. In the present study with 3.5 years of follow-up, 66.7% of PACG eyes showed structural progression and 33.3% of PACG eyes showed VF progression. Our study revealed a slightly higher proportion of progression, which might be due to the employment of additional progression criteria based on SD-OCT, different VF progression criteria, and study design. Nevertheless, a substantial portion of PACG eyes showed progression. Moreover, PAC/PACS eyes also revealed structural deterioration.

Although progression after lens extraction was observed, studies have rarely reported the risk factor for postoperative progression in PACD. Based on our literature search, there was only one study that reported the factors associated with disease progression after phacoemulsification in patients with CACG [23]. Lee et al. reported a 25% progression rate in patients with CACG after cataract surgery with risk factors of low preoperative VFI and high postoperative IOP. However, their study included only VF- and IOP-related parameters, whereas our study analyzed more putative risk factors for progression, such as CCT, AL, AS OCT parameter, and RNFL thickness. Related to VF progression, lower RNFL thickness and higher postoperative IOP fluctuation were risk factors for progression. Structural deterioration is generally considered to be preceded by VF alteration in glaucoma [24]. In our study, there is a possibility that the eyes that progressed in their VF had already experienced structural damage and, subsequently, may have worsened in their VF.

IOP fluctuation (both diurnal and long-term) has been considered as a potential independent risk factor for glaucoma progression and might be a better predictor than the mean IOP [25,26,27,28,29,30]. Baskaran et al. reported that PACG and PAC eyes showed higher diurnal IOP fluctuation compared to PACS and normal subjects [27]. In addition, Cheung et al. reported that PACG eyes had greater IOP fluctuation than PAC/PACS eyes [31]. Especially in PACG patients, circadian IOP fluctuation was significantly associated with disease progression [28]. The results of our study also showed that IOP fluctuation in PACG was related to VF progression, which is in line with a previous study. However, most previous studies excluded eyes with performed lens extraction during follow-up. In this study, only eyes with PACD that had undergone lens extraction were analyzed, and it was confirmed that IOP fluctuation appears as a risk factor in PACG even after surgery.

There are no previous studies regarding risk factor analysis in eyes with PACS and PAC after lens extraction. Sihota et al. reported a long-term clinical course of PAC eyes after LPI [32]. Among 72 eyes with PAC, eight eyes (11.1%) developed a visual field defect after 4 years of follow-up. The progressor showed a narrower angle, higher baseline MD, and larger inter-visit IOP fluctuation. In our study, those parameters were not different between progressor and non-progressor. Since the lens extraction stabilized IOP more than LPI, the difference in IOP fluctuation may have been reduced after lens extraction [6]. In the present study, the differences between progressed and non-progressed eyes were CCT and preoperative RNFL thickness. In logistic analysis, preoperative RNFL thickness was the only significant risk factor for structural progression. It is well known that RNFL thickness measured by OCT after acute angle closure glaucoma (AACG) was initially thicker, followed by subsequent decreases over time [33,34,35]. The factor related to RNFL loss after AACG was a longer duration of symptoms before receiving treatment [35,36]. In this study, eyes with progression showed RNFL thinning right before surgery; however, IOP was not different with non-progressed eyes. Therefore, it is possible that the damage caused by increased IOP, which was unknown before, may have led to RNFL thinning, and that these structurally weak eyes were more likely to show progression after lens extraction. However, as we described in our previous study, none of the eyes showed development of a VF defect after lens extraction in the PAC/PACS group [13]. Hence, lens extraction should be performed before glaucomatous damage commences, i.e., in the PAC/PACS stage. Additionally, our results suggest that those PAC/PACS eyes that underwent lens extraction revealed structural progression in eyes with thinner RNFL, and, thus, those eyes with thinner RNFL need to be followed with caution.

CCT has been identified as a substantial risk factor in POAG, and it was also identified as a risk factor for progression in PACG in a small number of studies [37,38,39,40]. Hong et al. reported that patients with CACG with a thinner cornea have a greater risk for visual field progression, even if they maintain low IOP [41]. A correlation between CCT and glaucomatous progression in the PAC spectrum is an interesting issue. Since lower CCT underestimates IOP when measured by Goldmann applanation tonometry, the actual IOP with lower CCT is thought to be high, which may have affected VF progression [42]. Meanwhile, there is growing evidence of corneal property and optic disc vulnerability in POAG eyes [43,44]. Similarly to POAG, PACG eyes can also be affected by such structural weakness. In our study, in both the PACS/PAC and PACG groups the progressed eye, according to structure or VF, showed thinner CCT than the non-progressed eye. In logistic analysis, CCT was a borderline risk factor for VF progression in PACG eyes. Therefore, thin CCT could be considered as a potential risk factor for glaucoma progression in PACD even after lens extraction; however, this issue warrants further investigation.

There are some limitations to our study. First, the study had a small sample size in each group. Due to small sample size, we compared the eyes with and without glaucomatous optic neuropathy in PACD. However, PACS and PAC may have different characteristics regarding progression. In addition, an average of 3.5 years of follow-up would not be enough to detect glaucomatous progression in some slow progressors. In this study, the PACS/PAC group did not show VF progression; however, the result might be different if we had a longer period of observation. Further study is needed, including a larger sample in each group and a longer follow-up period. Second, due to the retrospective nature of the study, there was no definite and uniform indication of lens extraction. There are many reasons for determination of lens extraction in angle closure eyes, and these reasons may co-exist in the same eye. Therefore, it would be difficult to determine operating criteria. Hence, a larger, randomized clinical trial may overcome these limitations.

## 5. Conclusions

In conclusion, eyes with PACD could progress, even after lens extraction. Preoperative thin RNFL was a risk factor for structural progression in PACS/PAC eyes. Preoperative thin RNFL and higher postoperative IOP fluctuation in PACG were risk factors for VF progression. Thin central cornea was a potential risk factor for the PAC spectrum. Our results may provide useful information for guiding clinicians on the monitoring of patients with PACD. Eyes having those features may require regular follow-ups for glaucoma progression, even with lowered IOP after lens extraction.

## Figures and Tables

**Table 1 jcm-11-02557-t001:** Comparison of clinical and demographic characteristics of eyes with primary angle closure suspect, primary angle closure, and primary angle closure glaucoma.

	Total (*n* = 77)	PACS/PAC (*n* = 41)	PACG (*n* = 36)	*p* Value
Age (year)	66.0 ± 7.6	65.1 ± 8.8	67.1 ± 5.7	0.222
Sex (M/F)	16/61	9/32	7/29	0.787 *
Baseline VA (logMAR)	0.14 ± 0.2	0.12 ± 0.3	0.15 ± 0.1	0.714
SE (diopter)	0.64 ± 1.6	0.57 ± 1.9	0.71 ± 1.2	0.708
Axial length (mm)	22.5 ± 0.8	22.5 ± 0.9	22.6 ± 0.7	0.498
CCT (μm)	544.0 ± 37.1	551.0 ± 32.4	536.1 ± 40.9	0.079
IOP (mmHg)	23.1 ± 14.4	23.6 ± 16.0	22.5 ± 12.5	0.763
ACD (mm)	1.9 ± 0.4	1.8 ± 0.4	2.0 ± 0.3	0.069
Lens vault (μm)	1154.0 ± 323.6	1209.0 ± 350.4	1091.4 ± 281.9	0.107
AOD500 (mm)	0.135 ± 0.09	0.126 ± 0.07	0.146 ± 0.09	0.344
Preoperative average RNFL thickness (μm)	88.4 ± 18.9	97.2 ± 15.5	78.3 ± 17.5	**<0.001 ^†^**
Preoperative VF MD (dB)	−4.88 ± 6.6	−2.13 ± 2.6	−8.01 ± 8.2	**<0.001**
Preoperative VF VFI (%)	88.5 ± 20.1	96.9 ± 6.2	78.9 ± 25.6	**<0.001**
Preoperative acute attack	23/77 (29.9%)	13/41 (31.7%)	10/36 (27.8%)	0.707
Time from diagnosis to surgery (year)	2.5 ± 2.8	2.5 ± 2.5	2.5 ± 3.2	0.996
Follow-up time after surgery (year)	3.5 ± 1.4	3.6 ± 1.4	3.5 ± 1.4	0.678
Postoperative mean follow-up IOP (mmHg)	13.4 ± 2.1	13.4 ± 1.8	13.5 ± 2.4	0.876
Glaucoma medication number	0.48 ± 0.6	0.15 ± 0.4	0.86 ± 0.7	**<0.001**
Last average RNFL thickness (μm)	82.60 ± 18.2	93.22 ± 11.2	70.50 ± 17.0	**<0.001 ^†^**
Last VF MD (dB)	−5.17 ± 7.3	−2.02 ± 2.1	−8.75 ± 9.2	**<0.001 ^†^**
Last VF VFI (%)	86.1 ± 23.3	96.6 ± 4.1	74.2 ± 29.8	**<0.001 ^†^**
Structure progression (%)	31/77 (40.3%)	7/41 (17.0%)	24/36 (66.7%)	**<0.001 ***
VF progression (%)	12/77 (15.6%)	0/41 (0%)	12/36 (33.3%)	**<0.001 ***

PACS = primary angle closure suspect; PAC = primary angle closure; PACG = primary angle closure glaucoma; M = male; F = female; VA = visual acuity; logMAR = logarithm of the minimal angle of resolution; SE = spherical equivalent; CCT = central corneal thickness; IOP = intraocular pressure; ACD = anterior chamber depth; AOD = angle opening distance; RNFL = retinal nerve fiber layer; VF = visual field; MD = mean deviation; VFI = visual field index; F/U = follow-up; * = chi-squared test; ^†^ = *p* < 0.05 by independent *t*-test unless otherwise indicated.

**Table 2 jcm-11-02557-t002:** Comparison of demographics and clinical characteristics in cases with and without progression of structure with primary angle closure suspect and primary angle closure.

	Non-Progressor (*n* = 34)	Progressor (*n* = 7)	*p* Value
Age (year)	64.6 ± 8.8	67.1 ± 9.0	0.747
Sex (M/F)	8/26	1/6	0.591 *
Base VA (logMAR)	0.14 ± 0.3	0.07 ± 0.1	0.879
Axial length (mm)	22.5 ± 0.9	22.5 ± 0.9	0.986
CCT (μm)	555.6 ± 32.6	528.6 ± 21.2	**0.049 ^†^**
IOP (mmHg)	24.3 ± 17.0	19.7 ± 10.1	0.722
ACD (mm)	1.8 ± 0.5	1.9 ± 0.2	0.282
Lens Vault (μm)	1227.9 ± 372.1	1117.1 ± 213.4	0.486
AOD500 (mm)	0.131 ± 0.08	0.105 ± 0.03	0.623
Preoperative average RNFL thickness (μm)	99.82 ± 15.1	84.57 ± 11.0	**0.001 ^†^**
Preoperative VF MD (dB)	−2.16 ± 2.6	−1.99 ± 2.4	0.808
Preoperative VF VFI (%)	96.6 ± 6.7	98.1 ± 2.0	0.663
Preoperative mean IOP (mmHg)	16.5 ± 3.5	16.1 ± 1.6	0.722
Preoperative peak IOP (mmHg)	27.0 ± 16.8	24.7 ± 9.8	0.623
Preoperative trough IOP (mmHg)	12.3 ± 2.7	12.4 ± 1.8	0.623
Preoperative IOP range (mmHg)	14.7 ± 17.8	12.2 ± 10.2	0.552
Preoperative LPI (N/Y)	12/22	3/4	0.705 *
Time from diagnosis to surgery (year)	2.6 ± 2.5	2.5 ± 2.8	0.933
Glaucoma medication number	0.12 ± 0.3	0.29 ± 0.5	0.258
Postoperative mean IOP (mmHg)	13.4 ± 1.7	13.4 ± 2.3	0.959
Postoperative peak IOP (mmHg)	16.0 ± 3.2	15.1 ± 2.7	0.599
Postoperative trough IOP (mmHg)	11.2 ± 1.6	11.4 ± 2.1	0.722
Postoperative IOP range (mmHg)	4.7 ± 2.8	3.7 ± 1.5	0.444

M = male; F = female; VA = visual acuity; logMAR = logarithm of the minimal angle of resolution; CCT = central corneal thickness; IOP = intraocular pressure; ACD = anterior chamber depth; AOD = angle opening distance; RNFL = retinal nerve fiber layer; VF = visual field; MD = mean deviation; VFI = visual field index; LPI = laser peripheral iridotomy; N = none; Y = yes; * = chi-squared test; = ^†^
*p* < 0.05 by Mann–Whitney test unless otherwise indicated.

**Table 3 jcm-11-02557-t003:** Comparison of demographics and clinical characteristics of progressed and non-progressed eyes with primary angle closure glaucoma.

Criteria	Structural Progression			Visual Field Progression		
	Non-Progressor (*n* = 12)	Progressor (*n* = 24)	*p* Value	Non-Progressor (*n* = 24)	Progressor (*n* = 12)	*p* Value
Age (year)	67.4 ± 7.8	67.0 ± 4.5	0.608	66.2 ± 6.6	68.9 ± 2.7	0.099 *
Sex (M/F)	3/9	4/20	0.551 ^†^	4/20	3/9	0.664 ^†^
Base VA (LogMAR)	0.18 ± 0.1	0.13 ± 1.3	0.166	0.16 ± 0.1	0.13 ± 0.1	0.934
Axial length (mm)	22.3 ± 0.7	22.7 ± 0.6	0.067 *	22.4 ± 0.5	22.8 ± 0.9	0.164 *
CCT (μm)	551.7 ± 40.0	528.3 ± 39.9	**0.041 ^‡^**	545.4 ± 41.5	517.5 ± 33.9	**0.026 ^‡^**
IOP (mmHg)	21.3 ± 11.0	23.2 ± 13.4	0.810	22.0 ± 12.7	23.6 ± 12.7	0.804
ACD (mm)	2.0 ± 0.2	2.0 ± 0.3	0.474 *	2.0 ± 0.3	2.1 ± 0.4	0.227 *
Lens vault (μm)	1159.2 ± 211.9	1057.5 ± 309.6	0.631	1145.8 ± 276.3	982.5 ± 271.4	0.102 *
AOD500 (mm)	0.121 ± 0.05	0.158 ± 0.11	0.398	0.142 ± 0.09	0.153 ± 0.13	0.934
Preoperative avg RNFL thickness (μm)	80.9 ± 16.3	77.0 ± 18.3	0.536 *	87.8 ± 17.7	69.9 ± 16.0	**0.028 ***
Preoperative VF MD (dB)	−10.09 ± 10.0	−6.97 ± 7.11	0.349 *	−6.6 ± 7.7	−10.9 ± 8.7	0.072
Preoperative VF VFI (%)	71.3 ± 29.9	83.7 ± 22.1	0.267	84.2 ± 23.4	68.5 ± 27.7	**0.024 ^‡^**
Preoperative mean IOP (mmHg)	16.7 ± 3.9	16.5 ± 5.4	0.608	15.7 ± 2.6	18.4 ± 7.6	0.585
Preoperative peak IOP (mmHg)	22.0 ± 10.5	24.9 ± 12.7	0.497	23.3 ± 12.3	25.4 ± 11.8	0.456
Preoperative trough IOP (mmHg)	12.9 ± 2.6	12.6 ± 6.2	0.273	12.0 ± 2.8	14.1 ± 8.3	0.753
Preoperative IOP range (mmHg)	9.1 ± 10.6	12.2 ± 13.5	0.297	11.3 ± 13.5	11.3 ± 11.1	0.704
Preoperative LPI (N/Y)	8/4	10/14	0.157 ^†^	11/13	7/5	0.480 ^†^
Time from diagnosis to surgery (year)	2.7 ± 4.5	2.4 ± 2.4	0.420	3.1 ± 3.8	1.4 ± 1.3	0.251
Glaucoma medication number	1.17 ± 0.8	0.71 ± 0.6	0.078	0.91 ± 0.7	0.75 ± 0.6	0.517
Postoperative mean IOP (mmHg)	12.9 ± 1.6	13.7 ± 2.7	0.608	13.3 ± 2.3	13.8 ± 2.8	0.753
Postoperative peak IOP (mmHg)	15.0 ± 1.8	16.5 ± 3.9	0.133 *	15.1 ± 2.4	17.8 ± 4.6	0.097 ^‡^
Postoperative trough IOP (mmHg)	10.2 ± 1.9	11.3 ± 3.0	0.270	11.1 ± 2.9	10.8 ± 2.6	0.631
Postoperative IOP range (mmHg)	4.7 ± 2.3	5.2 ± 3.5	0.905	4.0 ± 2.1	7.0 ± 4.0	**0.013 ^‡^**

M = male; F = female; VA = visual acuity; logMAR = logarithm of the minimal angle of resolution; CCT = central corneal thickness; IOP = intraocular pressure; ACD = anterior chamber depth; AOD = angle opening distance; RNFL = retinal nerve fiber layer; VF = visual field; MD = mean deviation; VFI = visual field index; LPI = laser peripheral iridotomy; N = none; Y = yes; * = unpaired *t*-test; ^†^ = chi-square test; ^‡^ = *p* < 0.05 by Mann–Whitney test unless otherwise indicated.

**Table 4 jcm-11-02557-t004:** Cox proportional hazards model to determine clinical variables associated with structure progression in primary angle closure suspect and primary angle closure.

	Univariate Analysis			Multivariate Analysis		
	HR	95% CI	*p* Value	HR	95% CI	*p* Value
Age	1.029	0.94–1.13	0.548			
Group (PACS as control)	0.552	0.12–2.47	0.437			
Axial length	0.965	0.40–2.30	0.936			
CCT	0.971	0.94–1.00	0.066	0.980	0.95–1.01	0.216
Baseline IOP	0.980	0.92–1.04	0.502			
RNFL thickness (preoperative)	0.928	0.88–0.97	**0.002**	0.928	0.88–0.97	**0.002**
VF MD (preoperative)	0.866	0.62–1.21	0.400			
VFI (preoperative)	0.914	0.81–1.03	0.130			
ACD	1.134	0.25–5.15	0.870			
Lens vault	1.000	0.99–1.00	0.843			
AOD500	0.001	0.00–233.0	0.275			
Preoperative LPI (not performed as control)	0.562	0.12–2.59	0.459			
Preoperative acute attack (did not happen as control)	0.913	0.18–4.72	0.913			
Mean IOP (preoperative)	0.995	0.76–1.30	0.974			
Mean IOP (postoperative)	0.967	0.63–1.48	0.878			

PACS = primary angle closure suspect; HR = hazards ratio; CI = confidence interval; CCT = central corneal thickness; IOP = intraocular pressure; RNFL = retinal nerve fiber layer; VF = visual field; MD = mean deviation; VFI = visual field index; ACD = anterior chamber depth; AOD = angle opening distance; LPI = laser peripheral iridotomy.

**Table 5 jcm-11-02557-t005:** Cox proportional hazards model to determine clinical variables associated with structure progression in primary angle closure glaucoma.

	Univariate Analysis		
	HR	95% CI	*p* Value
Age	1.023	0.95–1.10	0.531
Axial length	1.283	0.76–2.15	0.346
CCT	0.992	0.98–1.00	0.102
Baseline IOP	1.023	0.99–1.05	0.133
RNFL thickness (preoperative)	0.994	0.97–1.02	0.611
VF MD (preoperative)	1.023	0.96–1.09	0.475
VFI (preoperative)	1.007	0.99–1.03	0.486
ACD	1.671	0.42–6.58	0.463
Lens vault	0.999	0.99–1.00	0.515
AOD 500	8.330	0.08–924.2	0.378
Preoperative LPI (not performed as control)	0.968	0.43–2.20	0.939
Preoperative acute attack (did not happen as control)	1.280	0.54–3.00	0.571
Mean IOP (preoperative)	1.080	0.99–1.18	0.098
Mean IOP (postoperative)	1.101	0.94–1.29	0.237

HR = hazards ratio; CI = confidence interval; CCT = central corneal thickness; IOP = intraocular pressure; RNFL = retinal nerve fiber layer; VF = visual field; MD = mean deviation; VFI = visual field index; ACD = anterior chamber depth; AOD = angle opening distance; LPI = laser peripheral iridotomy.

**Table 6 jcm-11-02557-t006:** Cox proportional hazards model to determine clinical variables associated with visual field progression in primary angle closure glaucoma.

	Univariate Analysis			Multivariate Analysis		
	HR	95% CI	*p* Value	HR	95% CI	*p* Value
Age	1.105	0.98–1.25	0.117			
Axial length	1.669	0.78–3.55	0.184			
CCT	0.998	0.98–1.00	0.094			
Baseline IOP	1.014	0.98–1.06	0.478			
RNFL thickness (preoperative)	0.950	0.92–0.99	**0.006**	0.964	0.93–0.99	**0.043**
VF MD (preoperative)	0.945	0.89–1.01	0.079			
ACD	2.963	0.45–19.41	0.257			
Lens vault	0.998	0.99–1.00	0.104			
AOD500	2.803	0.01–1674.5	0.752			
Preoperative LPI (not performed as control)	0.494	0.16–1.58	0.233			
Preoperative acute attack (did not happen as control)	0.817	0.22–3.03	0.763			
Postoperative mean IOP	1.111	0.89–1.39	0.353			
Postoperative IOP range	1.353	1.13–1.62	**0.001**	1.296	1.06–1.58	**0.011**

HR = hazards ratio; CI = confidence interval; CCT = central corneal thickness; IOP = intraocular pressure; RNFL = retinal nerve fiber layer; VF = visual field; MD = mean deviation; VFI = visual field index; ACD = anterior chamber depth; AOD = angle opening distance; LPI = laser peripheral iridotomy.

## Data Availability

The datasets generated and analyzed during the current study are available from the corresponding author on reasonable request.

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
