# Peer review of "Factors Associated with Deterioration of Primary Angle Closure after Lens Extraction"

_jcm, 2022, doi:10.3390/jcm11092557_

Round 1

Reviewer 1 Report

I read with great interest this manuscript titled ‘’Factors associated with deterioration of primary angle closure after lens extraction’’ where authors studied some factors associated with glaucomatous deterioration in eyes with PAC after lens extraction. I found a very interested, well written and well done manuscript where authors also recognize its limitations, such as the low sample size and the retrospective nature of it. I would like to congratulate for the work.

Few comments:

I would like to recommend for future studies to try to predict a cut off for a lens extraction according to the preoperative RNFL thickness since according to the authors, that’s was the main risk factor for progression after the surgery and only 7 eyes in the PACS/PAC group showed a progression (only structural not in VF). Therefor I really believe that trying to find a cut off with a well-done ROC curves to know when is the best time to operate this patient should be important for our practice

I would also recommend for future studies not only use the Goldmann applanation tonometry but also new ones as CORVIS since the biomechanics has an influence in these patients and cand be also useful to analyze the glaucoma tendency with this tool.

Author Response

  • Thank you for excellent suggestion. Our team is planning a large-scaled prospective study to complement the limitations of this study. In the next study, we would try to analyze what you pointed out.
  • Thank you for your comment. We really agree with the importance of the corneal biomechanical properties in IOP measurement. Therefore, studies using tonometer that reflects corneal biomechanical characteristics will be needed in the future study.

Reviewer 2 Report

The authors explored factors related to glaucomatous progression in eyes with PACS, PAC, and PACG. Although the data were collected in a retrospective manner, the folow-up period was relatiely long (2 years) and the results were very helpful for the clinicians. I have only minor comment.
#1. The authors metion that higher postoperative IOP fluctuation is a risk factor for glaucomatous progression. How to define IOP fluctuation in this study? Also, are there any problems associated with cataract extraction? Are there any differences in IOP lowering treatment (adding eye drops or LI etc. )? Please describe more details about IOP fluctuation and add limitations if  the other related factors can be confounded the results?   

#2. In Table 2 and 3, it seems that there is no relation between IOP and glaucomatous progression, meaning that IOP itself may not be related to RNFL thinning in this type of glaucoma. If so, why the IOP fluctuation influences the progression? Please add the explanation. 

Author Response

  • Thank you for your comment. IOP fluctuation was calculated as the difference between the highest and lowest value of IOP among different visits. Previous publications provided several methods of calculating long-term IOP fluctuations: 1) Mean IOP; 2) standard deviation of IOP; 3) peak IOP; 4) IOP difference; all were related to short-term diurnal fluctuations, so we thought our calculation method was appropriate. (Tojo N, Abe S, Miyakoshi M, Hayashi A. Correlation between short-term and long-term intraocular pressure fluctuation in glaucoma patients. Clin Ophthalmol. 2016;10:1713-1717. Published 2016 Sep 2. doi:10.2147/OPTH.S116859). The method has been clarified in “Method” section in the line 147.
  • We included only non-complicated cataract surgery cases, so it was not be the confounding factor. Also, we excluded additional glaucoma surgery cases and did not perform LPI for treatment to lower IOP after lens extraction. Therefore, there would be no confounding factors according to the treatment method, because we included only cases with using IOP-lowering eyedrops after lens extraction.

Reviewer 3 Report

This is a nice paper and well written. How ever few points should be adders.

I could not find the change in ACD after the surgery. This value might be very important.

Another point is may be to normalized IOP after surgery to the CCT.

Author Response

  • Thank you for your comment. Unfortunately, we did not perform postoperative AS-OCT exams in all eyes. It has been reported that ACD increases, and angle widens after cataract surgery in previous several studies. Therefore, we did not routinely take AS-OCT after lens extraction. However, as you pointed out, there is a possibility that the change value of ACD may also affect the progression in eyes with PACD. Our team in planning a large-scale prospective study to complement the limitations of this study. That study will include postoperative ACD measurements.
  • ‘Another point is may be to normalized IOP after surgery to the CCT.’ Sorry, we could not understand this sentence.

Reviewer 4 Report

Correct in the line 160 and 161: "16 eyes were PACS, 24 eyes were PAC" - 16+24= 40. It should be together 41 eyes according Table 1.  

Author Response

  • Thank you for pointing out. We checked the patient data again and confirmed that the PACS group consisted of 17 eyes. We fixed it in the line 161.
